# Estimation of enteric methane emissions in dairy cows under grazing a silvopastoral system and a grass monoculture in the Colombian Amazonian foothills

Juan Pablo Narváez-Herrera[1], Joaquín Angulo-Arizala[1],
Wilson Andrés Barragán-Hernández[2], Liliana Mahecha-Ledesma[1]*

1 Universidad de Antioquia, Facultad de Ciencias Agrarias, Grupo de Investigación en Ciencias Agrarias (GRICA), Medellín, Colombia, 2 Corporación Colombiana de Investigación Agropecuaria (AGROSAVIA), Centro de Investigación El Nus, San Roque, Colombia

☉ These authors contributed equally to this work.
* liliana.mahecha@udea.edu.co

## Abstract

Mitigating enteric methane in the humid tropics, particularly in the Colombian Amazonian foothills, remains challenging due to limited field-based data under real grazing conditions. This study evaluated the performance of a laser methane detector (LMD) as a non-invasive alternative to traditional techniques, providing the first field-based validation of this approach in Amazonian grazing systems. Two contrasting production systems were compared: a silvopastoral system (SPS) with trees and shrubs, and a grass monoculture (traditional pasture, TP). A crossover design (two groups of five cows) was implemented across four periods. The LMD enabled repeated, activity measurements without disrupting natural behavior, capturing emissions during grazing, ruminating, resting, and milking. Daily $CH_4$ emissions were significantly lower in SPS than TP ($233 \pm 6.95$ vs. $277 \pm 8.87$ g $CH_4$ animal$^{-1}$ day$^{-1}$; $p < 0.0001$). Methane intensity also decreased in SPS when expressed per kg milk (15.5 vs. 20.7 g $CH_4$ kg$^{-1}$), energy-corrected milk (16.0 vs. 21.2 g $CH_4$ kg$^{-1}$), and dry matter intake (18.9 vs. 26.7 g $CH_4$ kg DMI$^{-1}$; all $p < 0.0001$). Classification was based on animal activity rather than diet, allowing detailed behavioral associations with $CH_4$ release dynamics. While the LMD requires strict environmental protocols and does not capture continuous 24-h data, its portability and non-invasive nature make it a practical, scalable tool for tropical field conditions. These results provide novel evidence supporting SPS as a mitigation strategy, strengthen GHG inventories in tropical livestock systems, and offer guidance for policymakers promoting sustainable production systems.

**Data availability statement:** All relevant data are within the manuscript and its Supporting Information files.

**Funding:** JPNH received funding from the Ministry of Science, Technology, and Innovation of Colombia through the Bicentennial Excellence Doctoral Scholarship Program - Cohort II (Doctoral Scholarship Agreement No. 20230030-20-21). The funders had no role in study design, data collection and analysis, decision to publish, or preparation of the manuscript.

**Competing interests:** The authors have declared that no competing interests exist.

## 1. Introduction

Livestock production is a key sector for global food security, but it is also one of the main sources of greenhouse gas (GHG) emissions, particularly methane ($CH_4$) generated through enteric fermentation in ruminants. It is estimated that livestock accounts for approximately 44% of global agricultural $CH_4$ emissions [1]. Methane has a global warming potential 28 times greater than that of carbon dioxide ($CO_2$) over a 100-year time horizon [2], which underscores the urgent need for accessible quantification methods and effective mitigation strategies.

Methanogenesis in the rumen is a microbial process primarily mediated by methanogenic archaea, which convert hydrogen and $CO_2$ into $CH_4$, with minor contributions from methylated substrates [3]. The resulting methane is predominantly released into the atmosphere via eructation [4]. Multiple factors influence $CH_4$ production, including diet, which alters hydrogen availability and fermentation patterns; ruminal microbiota composition, particularly the abundance of methanogens; and host genetics, which may affect rumen physiology and retention time [5].

Traditionally, $CH_4$ emissions from ruminants have been quantified using respiration chambers, the sulfur hexafluoride ($SF_6$) tracer technique, and portable systems such as Green Feed [6]. Respiration chambers, considered the reference method, offer high accuracy but are invasive, restrict natural behavior, and may induce stress-related artifacts due to confinement [7]. The $SF_6$ tracer technique enables long-term, individual measurements under grazing conditions and has proven valuable in field studies [8]. However, it requires ruminal infusion of permeation tubes and frequent collection of breath samples, which poses logistical and animal welfare challenges. In addition, $SF_6$ is an extremely potent greenhouse gas, with a 100-year global warming potential of approximately 25,200 relative to $CO_2$ when climate–carbon feedbacks are included [9]. Although the quantities released in tracer studies are minimal (on the order of micrograms per animal per day), the precautionary principle highlights the importance of considering its environmental implications. Portable systems such as Green Feed allow for repeated, non-invasive measurements in free-ranging animals but require training and concentrate feeding, which may alter grazing patterns and bias estimates [10]. As Green Feed captures emissions only during feeding bouts, daily $CH_4$ outputs are extrapolated, introducing additional uncertainty. Collectively, these limitations underscore the need for less intrusive, more representative, and scalable measurement technologies.

In recent years, the laser methane detector (LMD) has emerged as a viable alternative for non-invasive, real-time measurement of $CH_4$ emissions under field conditions [11]. Previous studies have validated its application in housed and grazing cattle [12–15], and activity-resolved field measurements have been reported [16], although variability related to sensor positioning, distance, and ambient conditions has also been noted [17]. Evidence from humid tropical systems remains scarce, which underscores the novelty of testing the LMD under Amazonian grazing conditions.

In tropical regions such as the Colombian Amazonian foothills, livestock production is predominantly based on extensive grazing, with increasing interest in silvopastoral systems (SPS). Traditional grass monocultures (*Urochloa* spp.) often exhibit low

crude protein, high fiber content, and marked seasonal fluctuations, limiting animal performance [18]. In traditional pastures, these constraints reduce intake and feed efficiency, and may exacerbate environmental intensity per unit of product. In contrast, SPS integrate trees, shrubs, and grasses, improving feed quality, stabilizing year-round availability [19]. Nevertheless, SPS also involve higher establishment costs and labor demands, which may constrain adoption by smallholders [20]. These systems have demonstrated potential for mitigating enteric $CH_4$ emissions by improving diet quality and nutrient conversion efficiency in ruminants [21].

The inclusion of tree and shrub species containing secondary metabolites, such as tannins, saponins, essential oils, and alkaloids, may modulate ruminal fermentation through different mechanisms. Tannins can bind to proteins and bacterial membranes, altering enzymatic activity and nutrient uptake, which reduces hydrogen availability for methanogenesis [22,23]. Saponins can decrease protozoal populations, thereby limiting symbiotic methanogens. Essential oils and alkaloids may disrupt microbial membranes and shift fermentation pathways toward alternative hydrogen sinks, such as propionate production. Collectively, these effects contribute to lowering methanogenic activity and mitigating enteric $CH_4$ emissions [24].

This study evaluated the use of the LMD to estimate enteric $CH_4$ emissions in dairy cows grazing either a silvopastoral system or a grass monoculture in the Colombian Amazonian foothills. By focusing on tropical production conditions, the research aims to provide novel insights into the applicability of LMD as a field tool and to inform strategies for improving GHG inventories and mitigation policies in developing countries.

## 2. Materials and methods

### 2.1. Study site

The study was conducted in the village of Aguanegra, located in the rural area of Puerto Asís, in the department of Putumayo, Colombia (0°33′09″N, 76°30′55″W), at an altitude of 270 meters above sea level. The region has a humid tropical climate, with an average annual temperature of 29 °C, relative humidity of 86%, and mean annual precipitation of 3,355 mm. Soils in the study area are classified as Ultisols with an average pH of 4.89 [25]. The study area is located in an equatorial tropical rainforest with a Köppen climate classification of Af [26]. It is characterized by a mosaic of natural pastures and secondary forest patches, where the dominant livestock forages are *Urochloa decumbens* and *Urochloa humidicola*.

**2.1.1. Experimental animal, design and management.** The study employed a crossover experimental design [27] to evaluate two grazing systems: a silvopastoral system (SPS) and a traditional pasture (TP). The SPS combined *Urochloa decumbens* with multipurpose tree and shrub species (*Piptocoma discolor, Clitoria fairchildiana*, and *Guazuma ulmifolia*), while *Erythrina poeppigiana* was used as a living fence. The TP consisted of a monoculture of *U. decumbens* managed under extensive grazing.

A total of ten lactating crossbred cows (Bos taurus × Bos indicus), of undefined Holstein and Gyr lineage, were selected based on minimal variation in daily milk yield, milk fat and protein content, days in milk (DIM), parity number (PN), body weight (BW), and body condition score (BCS), and subsequently allocated into two homogeneous groups (S1 Table). The evaluation lasted 76 days, divided into four consecutive 19-day periods, each with 14 days of adaptation [8,28] and 5 days of measurements [2]. During the first period, one group was assigned to the SPS and the other to the TP; from the second to the fourth period, treatments were alternated.

All procedures involving animals were conducted in accordance with protocols approved by the Animal Ethics Committee for Experimental Procedures (CEEA) of the University of Antioquia, under approval number 0146 dated June 7, 2022, ensuring animal welfare.

Two grazing systems were evaluated (Table 1) a silvopastoral system (SPS) consisting of *Piptocoma discolor* planted in double rows spaced 6 m between alleys and 0.5 m between plants (4,200 plants/ha); scattered trees of *Clitoria fairchildiana* and *Guazuma ulmifolia* planted at 20 m × 20 m spacing (25 trees/ha); *Erythrina poeppigiana* used as live fencing

**Table 1. Nutritional composition of the species evaluated in the grazing systems.**

| Variable | Silvopastoral System (SPS) | | | Traditional Pasture (TP) | |
|---|---|---|---|---|---|
| | Supplement | *P. discolor* | *U. decumbens* | Supplement | *U. decumbens* |
| DM, % | 87 ± 2.98 | 25.70 ± 1.53 | 23.0 ± 0.98 | 87 ± 2.98 | 22.85 ± 0.68 |
| CP, % | 16.02 ± 0.52 | 25.01 ± 0.69 | 9.79 ± 0.48 | 16.02 ± 0.52 | 8.64 ± 0.26 |
| NDF, % | 15.56 ± 0.54 | 50.01 ± 1.50 | 59.74 ± 3.01 | 15.56 ± 0.54 | 69.81 ± 4.10 |
| ADF, % | 7.62 ± 0.25 | 24.72 ± 1.23 | 31.87 ± 1.83 | 7.62 ± 0.25 | 37.08 ± 1.54 |
| NEL, Mcal* | 1.89 ± 0.06 | 1.49 ± 0.06 | 1.24 ± 0.09 | 1.89 ± 0.06 | 1.10 ± 0.05 |
| TT, g/kg$^{-1}$ DM | – | 17.28 ± 0.05 | 0.08 ± 0.06 | – | 0.08 ± 0.06 |
| Sap, g/kg$^{-1}$ DM | – | 23.46 ± 0.07 | 1.10 ± 0.05 | – | 1.10 ± 0.05 |

DM = dry matter; CP = crude protein; NDF = neutral detergent fiber; ADF = acid detergent fiber; NEL = net energy for lactation. TT = Total Tannins; Sap = Saponins. *Adapted from Buxadé (1994): NEL = 0.677 × DE – 0.359; DE = digestible energy, kcal DE.

with 6 m spacing between individuals (66 trees/ha); and a ground cover of *Urochloa decumbens*. At the time of this study, the SPS had been established for 18 months, and the arboreal species (*C. fairchildiana*, *G. ulmifolia* and *E. poeppigiana*) were still in the early growth stages, without a developed canopy or tree-like structure. Consequently, the potential impact on microclimate or animal behavior was not considered in the analysis. It was hypothesized that only *P. discolor*, due to its shrubby architecture and high planting density, was capable of contributing structurally to the vegetation stratum.

The traditional pasture system (TP) was based on a monoculture of *Urochloa decumbens*, under extensive grazing with a low tree density (<25 trees/ha), primarily composed of pre-existing *Cordia alliodora* individuals (3–5 years old, 30–40 cm Diameter at Breast Height (DBH)), and managed under a 42-day rotational grazing cycle. While these trees were already established prior to the experiment, their presence reflects the typical structure of low-density arboreal cover in extensively managed tropical pastures, and no additional planting or modification was made for the purpose of this study. All animals received a daily supplement of 3 kg of commercial concentrate throughout the experimental period to ensure comparable baseline nutrition across treatments.

## 2.2. Milk yield and dry matter intake (DMI)

Milk yield and DMI data were obtained from a complementary study conducted under the same experimental conditions and using the same animals [20]. Milk production (kg/day) was determined individually through manual milking performed twice daily. The total volume was measured directly from the calibrated milking bucket immediately after each session. A standard milk density of 1.032 g/mL was used to convert volume to weight. Energy-corrected milk (ECM) was calculated according to [29], using the following equation:

$$ECM(kg) = production\ (kg/day) * [0.383 * milk\ fat\ \% + 0.242 * milk\ protein\ \% + 0.7832)/3.14 \tag{1}$$

Milk fat and protein contents were determined by infrared spectroscopy using a MilkoScan FT+ milk analyzer (Foss, Hillerød, Denmark).

Dry matter intake was estimated using two methods, the double marker method and the agronomic method. The double marker combining chromium oxide ($Cr_2O_3$) as an external marker and acid detergent lignin (ADL) as an internal marker. $Cr_2O_3$ was administered at a dose of 15 g/cow/day, and fecal samples were collected over five consecutive days per period. The chromium recovery rate was 79.8%. Intake partitioning between forage and supplement was estimated based on differential lignin content, following the procedure described by [30]. A detailed methodological description and the corresponding results are available in [20]. This method was used to estimate the total voluntary intake of forage. The agronomic method was used to determine the grass-to-shrub ratio in the forage diet

by measuring the biomass of grasses and shrubs before and after grazing in representative paddocks, following the double sampling technique described by [31] and adapted by [32]. Measurements were conducted during the adaptation period, and the relative proportion of each forage species was used to estimate their contribution to the total voluntary intake. To ensure comparable intake opportunities and a consistent grass-to-shrub ratio across periods, paddock area and stocking density were adjusted based on forage allowance, maintaining a target offer of 10–12% of body weight (kg DM/cow/day) [33,34].

## 2.3. Methane emission measurements

Methane emissions were measured using a laser methane detector (LMD; model LMm-G, Crowcon, Erlanger, KY, USA). The device operates on high-selectivity infrared spectroscopy targeting the $CH_4$ absorption band, following the recommendations of [12–15]. Measurements were performed by directing the laser beam at the animal's nostril region from a fixed distance of 2 m. Each session lasted 4 min, with data captured at 0.1-second intervals to ensure coverage of both respiration and eructation cycles. To minimize operator bias, all measurements were performed by the same trained operator.

Within each 5-day evaluation window, measurements were conducted on the first two days that met pre-defined environmental quality criteria (no rainfall, wind speed $\leq 2$ m s$^{-1}$, and stable wind direction); if criteria were not met, measurements were postponed to the next day within the window. Each animal was then evaluated twice per qualifying day during the morning (06:00–10:30) and afternoon (16:30–19:00) milking routines. Concurrently, ambient temperature, relative humidity, and wind speed were recorded with a portable weather station (Davis Vantage Pro2, Hayward, CA, USA). Wind direction relative to the detector was classified as headwind, crosswind, or tailwind according to [13]. Data were collected only when wind speed was $\leq 2$ m/s to ensure stable detection conditions. Background $CH_4$ concentrations were measured 2 m upwind of the grazing area before and after each session and were subtracted from animal-level signals to correct for environmental sources.

Methane signal processing followed the dynamic thresholding approach described by [11,15]. For each animal and session, the arithmetic mean plus one standard deviation of the $CH_4$ signal was defined as the threshold. Peaks above this value were classified as eructation events, while those below were attributed to respiration. Average $CH_4$ concentrations for respiration (R_CH$_4$), eructation (E_CH$_4$), and overall mean (MEAN_CH$_4$) were then obtained. Concentrations (ppm × m) were converted to grams of $CH_4$ per day per kilogram of body weight using the equation of [15]:

$$CH_4 \left( g \, day^{-1} \right) = mean \; CH_4 \; x \; V \; x \; R \; x \; \propto \; x \; \beta \; x \; 10^6 \; x \; 1440$$

In which V is the tidal volume (3800 mL), R is the respiratory rate (respiratory peaks), $\propto$ is the conversion factor of $CH_4$ production from mL to g (0.000667 g/mL), β is the correction factor for the difference between breath and total $CH_4$ production. The estimation of daily $CH_4$ emission was normalized to the daily milk production of each cow.

## 2.4. Animal behavior recording

Animal behavior was recorded concurrently with LMD sessions using focal animal, continuous sampling [35]. Activities were classified as grazing (PST), ruminating (RM), resting (DE), idle (OC), and milking (OR). Each cow was observed for the full LMD session ($\geq$240 s), twice daily on two of the five evaluation days (4 sessions/cow), within the 06:00–10:30 and 16:30–19:00 windows. Two trained observers participated: one operated the LMD and a second logged behavior with a stopwatch, time-stamping every change of state. Observers stood 2–3 m away, outside the flight zone; disturbances were annotated and affected sessions repeated. For each session, the proportion of time per activity was calculated and matched to the corresponding $CH_4$ series.

## 2.5. Temperature-Humidity Index (THI)

Ambient temperature (T, °C) and relative humidity (RH, %) were recorded daily using a portable weather station (Davis Vantage Pro2, Hayward, CA, USA) located at the experimental site. These data were used to calculate the daily Temperature-Humidity Index (THI) following the equation proposed by the National Research Council [36]:

$$THI = (1.8 \ x \ T + 32) - [(0.55 - 0.0055 \ x \ HR)x \ (1.8 \ x \ T - 26)]$$

Where T is the ambient temperature in °C and RH is the relative humidity in %.

## 2.6. Data processing and statistical analysis

The analysis of the data was conducted using a generalized linear mixed model (GLMM) fitted with the glmmTMB package [37] in R software version 4.3.1 (R Core Team). A log link function was specified for the response variables (daily methane emission, methane yield, and methane intensity), to correct for their positive skewness and satisfy model assumptions. The fixed effect was the grazing system (treatment) with two levels: silvopastoral system (SPS) and traditional pasture (TP), and the temperature–humidity index (THI) was included as a continuous covariate. Period, animal group, and individual cow were modeled as random effects.

$$y_{ijkl} = \mu + T_i + \beta + P_j + G_k + C_l + \varepsilon_{ijkl}$$

Where Y is the response variable, $\mu$ is the overall mean, $T_i$ is the fixed effect of treatment (i = SPS, TP), $\beta$ is the regression coefficient for the covariate THI, $P_j$, $G_k$, and $C_l$ are the random effects of period (j = 1–4), animal group (k = G1, G2), and cow (individual subject), respectively, with $C_l \sim N(0, \sigma\_c^2)$, and $\varepsilon_{ijkl}$ is the residual error term. Model diagnostics were performed using the DHARMa package [38], based on simulation of scaled residuals. Adjusted means (LSMeans) for the treatment effect were estimated using the emmeans package [39], and pairwise comparisons were performed using the Tukey test. Statistical significance was declared at p < 0.05.

## 3. Results

### 3.1. Milk yield and dry matter intake

Cows managed under the silvopastoral system (SPS) had significantly higher milk yield and dry matter intake (DMI) compared to those in the traditional pasture (TP) (Table 2). These differences were statistically significant (p < 0.05) and were used as the basis for calculating methane yield indicators. In the SPS, a grass-to-shrub ratio of 88:12 was observed in the grazed biomass, with an estimated average intake of *Piptocoma discolor* of 1.55 kg DM/animal/day. This value was calculated based on the proportional contribution of the shrub to the total forage biomass, as determined using the agronomic method.

**Table 2. Daily dry matter intake (DMI) and milk yield in dairy cows grazing in a silvopastoral system (SPS) and a traditional pasture (TP).**

| Treatment | DMI (kg/animal/day) | Milk yield (kg/animal/day) |
|---|---|---|
| SPS | 12.9±0.12 | 14.13±0.80 |
| TP | 10.5±0.12 | 12.92±0.80 |
| p-value | 0.001 | 0.001 |

SPS: silvopastoral system; TP: traditional pasture; DMI: dry matter intake. Values are means±SD for 10 cows measured across four periods.

Although the tree species (*Clitoria fairchildiana*, *Guazuma ulmifolia*, and *Erythrina poeppigiana*) were present in the silvopastoral system (SPS) paddocks, they were not intended for animal consumption. Consequently, their contribution to the diet was negligible and thus excluded from intake estimates

### 3.2. Relationship between the Temperature-Humidity Index (THI) and methane emissions

The temperature-humidity index (THI) progressively increased across the four evaluation periods, ranging from 75.95 in the first period to 87.19 in the fourth period (Table 3). Mean THI values were consistently above 75 in all periods, with the highest values recorded during the last two periods. THI values were incorporated as a covariate in the statistical model to adjust for environmental variability in methane emissions.

### 3.3. Estimation of methane emissions using the LMD

Methane yield indicators reported significant differences between grazing systems (Table 4). Cows in the SPS emitted lower amounts of $CH_4$ per animal per day, as well as per kilogram of milk, energy-corrected milk (ECM), and dry matter intake (DMI), compared to those in the traditional pasture (TP). All differences were significant ($p < 0.05$).

### 3.4. Animal behavior and relationship with methane emissions

As shown in Fig 1, enteric $CH_4$ emissions varied according to behavioral activity in both grazing systems. The highest values were recorded during grazing, followed by ruminating, whereas the lowest emissions occurred during milking and resting. During grazing, cows in the traditional pasture (TP) emitted on average 321.2 g $CH_4$/animal/day, compared with 305.2 g $CH_4$/animal/day in the silvopastoral system (SPS). For resting periods, emissions averaged 190.2 g $CH_4$/animal/day in TP and 182.4 g $CH_4$/animal/day in SPS. Thus, TP showed numerically higher emissions, with values between 4

**Table 3. Temperature-Humidity Index (THI) and daily methane emissions in cows grazing SPS and TP systems.**

| Period | THI (± SD) | SPS CH$_4$ (g/animal/day) | TP CH$_4$ (g/animal/day) |
|---|---|---|---|
| 1 | 75.95 ± 1.54 | 206.13 ± 72.24 | 269.53 ± 95.87 |
| 2 | 79.85 ± 1.52 | 222.89 ± 15.03 | 280.56 ± 44.78 |
| 3 | 81.48 ± 1.47 | 248.03 ± 26.74 | 280.04 ± 42.93 |
| 4 | 87.19 ± 1.64 | 255.10 ± 38.62 | 278.47 ± 43.78 |

Values are expressed as mean ± standard deviation. THI: Temperature-Humidity Index. CH$_4$: methane. SPS: silvopastoral system. TP: traditional pasture. Values represent 10 cows measured over four periods.

**Table 4. Enteric methane emissions (total and intensity-based) in dairy cows grazing a silvopastoral system (SPS) or a traditional pasture (TP) in the Colombian Amazonian foothills.**

| Treatment | CH$_4$ (g/animal/day) | CH$_4$ (g/kg milk) | CH$_4$ (g/kg ECM) | CH$_4$ (g/kg DMI) |
|---|---|---|---|---|
| SPS | 233 ± 6.95 a | 15.5 ± 0.356 a | 16.0 ± 0.436 a | 18.9 ± 0.433 a |
| TP | 277 ± 8.87 b | 20.7 ± 0.584 b | 21.2 ± 0.659 b | 26.7 ± 0.744 b |
| SEM | 7.9658 | 1.1148 | 1.1279 | 0.7635 |
| *p*-value | 0.0001 | 0.0001 | 0.0001 | 0.0001 |

Values are means ± standard error of the mean (SEM). Different superscript letters in the same column indicate significant differences (Tukey test, $p < 0.05$). Adjusted means (LSMeans) were obtained from the GLMM; n = 10 cows measured across four periods. ECM = energy-corrected milk.

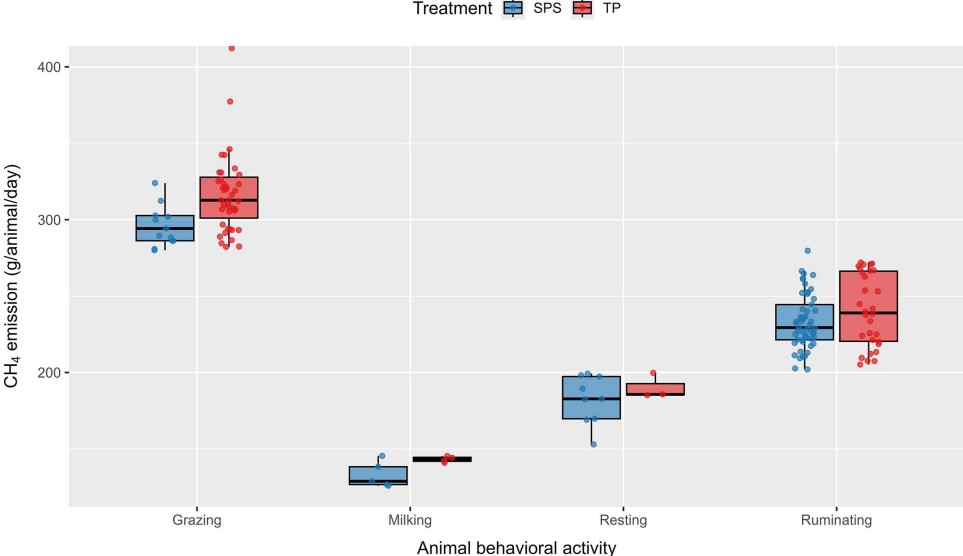

**Fig 1. Enteric CH₄ emissions by behavioral activity in silvopastoral (SPS) and traditional pasture (TP) systems.** The boxes represent the inter-quartile range and the central line indicates the median. No significant differences (p > 0.05) were detected between grazing systems for any behavioral activity.

and 5% (ratios of 1.04–1.05) above SPS across all observed activities; however, these differences were not statistically significant (p > 0.05).

## 4. Discussion

Quantification of enteric methane emissions using a Laser Methane Detector (LMD) enabled representative estimates under real grazing conditions. In this study, the LMD discriminated emissions between silvopastoral (SPS) and traditional pasture (TP) systems, capturing treatment-level contrasts under field conditions. Prior work has highlighted the LMD's portability, relatively low cost, and capacity for repeated, non-invasive measurements [40]. Nonetheless, technical limitations notably sensitivity to environmental factors such as wind speed, distance, and measurement angle can affect accuracy [40,41]. To ensure reliability despite these constraints, we implemented standardized protocols: a fixed detector–muzzle distance of 2 m, a minimum sampling duration of ≥ 240 s per session, and measurements restricted to wind speeds ≤ 2 m·s⁻¹ with documented wind direction [15,42]. Accordingly, inference emphasizes within-study contrasts (SPS vs. TP) under a harmonized protocol rather than cross-study comparison of absolute values.

Silvopastoral diets that combine higher-quality forages with moderate levels of functional secondary metabolites can lower enteric CH₄ by steering fermentation toward more glucogenic end-products and by constraining hydrogen availability for methanogenesis [43–45]. Consistent with this mechanism, SPS reduced daily CH₄ per animal by 15.9% relative to TP despite higher DMI, which aligns with the higher nutritive value of the SPS diet and the inclusion of *Piptocoma discolor*. The lower CH₄ per kg DMI in SPS indicates improved energy capture per unit feed, a response expected when diet quality increases intake without proportionally increasing methanogenesis because fermentation shifts toward propionate and bioactive compounds moderate hydrogen flux [46–48]. This pattern is also compatible with *in vitro* observations using woody forage plants such as *Tithonia diversifolia*, which reported reduced methane production relative to grass-only substrates [49].

Consistent with these diet-driven mechanisms, productivity responses mirrored the mitigation pattern. SPS also yielded higher milk production than TP, which helps explain the pronounced reduction in CH₄ per unit of milk and ECM. From an efficiency standpoint, greater milk output under similar management dilutes maintenance requirements, lowering GHG

intensity per unit product. The concordance between higher DMI and higher milk yield in SPS suggests improved energy partitioning toward lactation rather than gaseous losses [50], in line with the lower $CH_4$/kg milk and $CH_4$/kg ECM reported here. Importantly, both systems received the same concentrate allowance; therefore, observed differences are principally attributable to the forage base and botanical composition under grazing.

The findings are in accordance with [51], who observed emissions ranging from 207 to 228 g $CH_4$/animal/day in pasture-based systems with *Brachiaria humidicola* and 15% *Tithonia diversifolia* using polytunnels. Similarly, [1] reported 205 g $CH_4$/animal/day in Jersey cows under European pasture systems using the eddy covariance technique. By contrast, using the LMD and a respiratory-based equation incorporating tidal volume, respiratory rate, and standard conversion factors [12]. [15] reported 328.6 ± 160.0 g $CH_4$/animal/day in Mediterranean buffaloes; under controlled conditions and a 120-s sampling duration, [11] reported 53.9 g $CH_4$/animal/day in Jerseys and 60.7 g $CH_4$/animal/day in Holsteins. According to [42], although the LMD is sensitive to sampling duration, it yields reproducible estimates when measurement conditions are standardized; adopting ≥ 240 s per event, as implemented here, improves precision and comparability across sessions. However, as noted by [40], the absence of a fully standardized LMD protocol still limits inter-study comparability of absolute values.

Beyond absolute emissions, methane intensity is widely regarded as a more robust indicator of environmental efficiency because it accounts for productive output [52]. In the present study, SPS showed significantly lower $CH_4$ intensities per kilogram of milk, energy-corrected milk (ECM), and dry matter intake (DMI), reflecting more efficient nutrient use. Specifically, $CH_4$ per kilogram of DMI in SPS was reduced by 29% compared to TP, and $CH_4$ per kilogram of ECM was 25% lower, highlighting better feed conversion efficiency. These improvements are linked to the higher nutritional quality of the SPS forage base, which included *P. discolor*, characterized by lower fiber, higher crude protein (up to 27.5%), and greater energy availability (up to 1.52 Mcal/kg DM) during early regrowth [20]. This species also contains functional secondary metabolites such as tannins and saponins, which may modulate ruminal fermentation and improve nitrogen utilization efficiency. Similar variability in methane intensity across systems and metrics has been reported by [1], with values ranging from 5.4 to 12.47 g $CH_4$/kg ECM in Jersey cows under supplemented grazing. The higher intensities observed here are consistent with full grazing and limited concentrate input. Overall, the between-system contrast likely reflects the lower nutritional quality and higher fiber content (particularly ADF) of *Urochloa decumbens* in TP versus the higher-quality forage base in SPS, including *P. discolor*.

Comparatively, [53]reported 12.3 g $CH_4$/kg DMI in intensive systems with high supplementation, and [11] found even lower values (11.1 g $CH_4$/kg DMI) in Holstein cows under rotational grazing with concentrate. Similarly, [54] reported 16.1 g $CH_4$/kg DMI and 11.9 g $CH_4$/kg milk in high-yielding Holstein cows (27 kg/day) housed under confinement with *Cenchrus clandestinus* grass and concentrate fed to yield. These contrasts illustrate the impact of concentrate inclusion and controlled intake on fermentation efficiency. In our field setting, TP exhibited higher methane yield (26.7 g $CH_4$/kg DMI; 20.7 g $CH_4$/kg milk), whereas SPS showed a more efficient profile (18.9 g $CH_4$/kg DMI; 15.5 g $CH_4$/kg milk), consistent with the measured higher diet quality in SPS and the inclusion of *P. discolor*, which may enhance fermentation. [51] observed similar improvements with Tithonia diversifolia in tropical diets, reinforcing the role of woody species functional traits in mitigation.

Considered together, methane intensity and yield indicators suggest that SPS improves the environmental efficiency of dairy production systems not only by reducing total emissions but also by decreasing $CH_4$ per unit of milk and per kilogram of DMI. This finding is aligned with [55], who found that Holsteins in SPS emitted 246.7 g $CH_4$/animal/day (15.4% lower than 291.5 g $CH_4$/animal/day in TP). Lower intensities per unit output were also reported (13.7 vs. 23.8 g $CH_4$/kg fat-corrected milk; 14.1 vs. 18.6 g $CH_4$/kg DMI). The inclusion of tree/shrub species such as *Eucalyptus* sp., *Alnus acuminata*, *Acacia melanoxylon*, and *Sambucus peruviana* improved diet quality, increased FCM yield (19.1 vs. 12.3 kg/day), and reduced Ym (3.4% vs. 4.5%), indicating reinforce its potential as an integrated strategy to mitigate emissions in tropical livestock systems.

   

Enteric methane emissions are closely linked to behavioral activity, as variation in intake rate, fermentation dynamics, and digestive efficiency influence $CH_4$ output [56]. As reported by [57], cows with longer rumination times emitted 18% more $CH_4$ per day, partly due to increased intake. From a fermentative standpoint, greater intake of high-ADF forages such as *Urochloa decumbens* stimulates cellulolytic bacteria, favoring acetate production and elevated hydrogen release, thereby stimulating methanogenesis [47,58]. In contrast, SPS included shrubs with lower ADF, higher digestibility, and condensed tannins, which can suppress methanogenic archaea or modulate fermentation [45,58]. These attributes provide a coherent explanation for the lower methane yield in SPS even under higher intake.

In this study, higher emission rates were observed during grazing, particularly in TP, where $CH_4$ output exceeded SPS by 13.5%. This outcome is consistent with evidence that active forage intake increases fermentative $H_2$, the primary substrate for methanogenesis [59,60]. Grazing can also coincide with higher thermal load, increasing respiratory rate and ventilation, potentially raising the frequency of methane exhalation through greater tidal volume and gas exchange, as shown with spirometric monitoring under field conditions [61]. Methane during rumination showed greater variability, likely reflecting eructation of accumulated gases linked to the secondary reticulorumen contraction cycle [40,62]; despite reduced primary fermentation during rumination, measurable peaks persist [63]. Conversely, resting was associated with lower emissions, consistent with reduced fermentative activity and metabolic demand, in agreement with [64], who observed lower $CH_4$ rates, greater energy efficiency, and lower $CH_4$ energy losses per liter of milk in less active cows. Taken together, these activity-specific patterns are consistent with prior reports [65].

During the milking routine, $CH_4$ emissions were minimal. This activity represents a phase of low digestive activity, in which neither intake forage occurs, and where human interaction can influence respiratory patterns [64]. Some studies suggest that during milking or brief confinement, the respiratory pattern may become more irregular but does not significantly increase $CH_4$ concentration [60,62]. Given that the arboreal component in our SPS was at an early growth stage without developed canopy, microclimatic effects on emissions were likely limited during the study period, and any potential buffering should be interpreted cautiously until full canopy development is achieved [44,66]. Furthermore, the use of the LMD in behavioral studies represents a valuable tool to identify temporal and physiological patterns of enteric $CH_4$ emission. However, its accuracy depends critically on the standardization of the measurement protocol, as emphasized in recent methodological studies [40,42].

## 5. Conclusions

Silvopastoral systems (SPS) in the Colombian Amazonian foothills effectively reduce enteric methane intensity, thereby improving the environmental efficiency of dairy production in the humid tropics. This mitigation effect is consistent with the superior nutritional quality of the SPS diet, particularly the inclusion of *Piptocoma discolor* and its functional secondary metabolites. Methodologically, this study validates the Laser Methane Detector (LMD) as a practical, non-invasive tool for assessing emissions in grazing animals, capable of detecting differences related to both treatment and behavior. While these findings are robust, they are contextualized by the study's modest sample size and short duration, highlighting the need for further research. Overall, our results champion SPS as a viable strategy for sustainable dairy production. We recommend future work focus on longer evaluation periods and the continued standardization of LMD protocols to enhance cross-study comparability.

## Supporting information

**S1 Table. Characteristics of experimental animals.**
(DOCX)

**S2 Database. Raw data used for the statistical analyses of enteric methane emissions, milk production, dry matter intake, and behavioral activities.**
(XLSX)

                                                                     

## Acknowledgments

The authors express their gratitude to the owners of La Primavera Farm, Mr. Aquilino Giraldo and Ms. Luz Marina Mejia, their family.

## Author contributions

**Conceptualization:** Juan Pablo Narváez-Herrera , Joaquín Angulo-Arizala, Wilson Andrés Barragán-Hernández, Liliana Mahecha-Ledesma.

**Data curation:** Juan Pablo Narváez-Herrera.

**Formal analysis:** Juan Pablo Narváez-Herrera.

**Funding acquisition:** Juan Pablo Narváez-Herrera, Liliana Mahecha-Ledesma.

**Investigation:** Juan Pablo Narváez-Herrera, Wilson Andrés Barragán-Hernández.

**Methodology:** Juan Pablo Narváez-Herrera, Joaquín Angulo-Arizala, Wilson Andrés Barragán-Hernández, Liliana Mahecha-Ledesma.

**Project administration:** Liliana Mahecha-Ledesma.

**Software:** Wilson Andrés Barragán-Hernández.

**Supervision:** Joaquín Angulo-Arizala, Liliana Mahecha-Ledesma.

**Writing – original draft:** Juan Pablo Narváez-Herrera.

**Writing – review & editing:** Joaquín Angulo-Arizala, Wilson Andrés Barragán-Hernández, Liliana Mahecha-Ledesma.

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
