## [Decision Letter · Decision Letter 0]

11 Sep 2025

Dear Dr. Mahecha-Ledesma,

Thank you for submitting your manuscript to PLOS ONE. After careful consideration, we feel that it has merit but does not fully meet PLOS ONE’s publication criteria as it currently stands. Therefore, we invite you to submit a revised version of the manuscript that addresses the points raised during the review process.

Please submit your revised manuscript by Oct 26 2025 11:59PM. If you will need more time than this to complete your revisions, please reply to this message or contact the journal office at plosone@plos.org . A rebuttal letter that responds to each point raised by the academic editor and reviewer(s). You should upload this letter as a separate file labeled 'Response to Reviewers'.A marked-up copy of your manuscript that highlights changes made to the original version. You should upload this as a separate file labeled 'Revised Manuscript with Track Changes'.An unmarked version of your revised paper without tracked changes. You should upload this as a separate file labeled 'Manuscript'.

We look forward to receiving your revised manuscript.

Kind regards,

Susmita Lahiri (Ganguly)

Academic Editor

PLOS ONE

Journal Requirements:

[JPNH received funding from the Ministry of Science, Technology, and Innovation of Colombia through the Bicentennial Excellence Doctoral Scholarship Program - Cohort II (Doctoral Scholarship Agreement No. 20230030-20-21).].

4. Thank you for stating the following in your manuscript:

[To achieve the results of this research, various technical and financial funding sources were sought: The Ministry of Science, Technology, and Innovation of Colombia, the Bicentennial Excellence Doctoral Scholarship Program - Cohort II, Doctoral Scholarship Agreement No. 20230030-20-21, and doctoral study funds for Juan Pablo Narváez-Herrera. Additionally, this research received technical support from the GRICA Research Group – Sustainable Animal Production Systems Line at the University of Antioquia, as well as the National Learning Service (SENA) – Putumayo Regional Office.]

[JPNH received funding from the Ministry of Science, Technology, and Innovation of Colombia through the Bicentennial Excellence Doctoral Scholarship Program - Cohort II (Doctoral Scholarship Agreement No. 20230030-20-21).]

5. We note that your Data Availability Statement is currently as follows: [All relevant data are within the manuscript and its Supporting Information files.]

Please confirm at this time whether or not your submission contains all raw data required to replicate the results of your study. Authors must share the “minimal data set” for their submission. PLOS defines the minimal data set to consist of the data required to replicate all study findings reported in the article, as well as related metadata and methods (https://journals.plos.org/plosone/s/data-availability#loc-minimal-data-set-definition ).

If your submission does not contain these data, please either upload them as Supporting Information files or deposit them to a stable, public repository and provide us with the relevant URLs, DOIs, or accession numbers. For a list of recommended repositories, please see https://journals.plos.org/plosone/s/recommended-repositories .

6. PLOS requires an ORCID iD for the corresponding author in Editorial Manager on papers submitted after December 6th, 2016. Please ensure that you have an ORCID iD and that it is validated in Editorial Manager. To do this, go to ‘Update my Information’ (in the upper left-hand corner of the main menu), and click on the Fetch/Validate link next to the ORCID field. This will take you to the ORCID site and allow you to create a new iD or authenticate a pre-existing iD in Editorial Manager.

7. We note that you have referenced (Narváez-Herrera JP, Angulo-Arizala J, Barragán-Hernández WA, Riascos-Guerrero YM, Mahecha Ledesma L. From leaves to liters: Enhancing dairy efficiency through silvopastoral integration in the Amazonian foothills. Manuscript submitted for publication. 2025.) which has currently not yet been accepted for publication. Please remove this from your References and amend this to state in the body of your manuscript: (ie “Bewick et al. [Unpublished]”) as detailed online in our guide for authors.

8. Your ethics statement should only appear in the Methods section of your manuscript. If your ethics statement is written in any section besides the Methods, please delete it from any other section.

9. We note that Figure 1 in your submission contain copyrighted images. All PLOS content is published under the Creative Commons Attribution License (CC BY 4.0), which means that the manuscript, images, and Supporting Information files will be freely available online, and any third party is permitted to access, download, copy, distribute, and use these materials in any way, even commercially, with proper attribution. For more information, see our copyright guidelines: http://journals.plos.org/plosone/s/licenses-and-copyright.

Reviewers' comments:

Reviewer's Responses to Questions

**Comments to the Author**

1. Is the manuscript technically sound, and do the data support the conclusions?

Reviewer #1: Partly

Reviewer #2: Yes

Reviewer #3: Yes

Reviewer #4: Yes

2. Has the statistical analysis been performed appropriately and rigorously?

Reviewer #1: No

Reviewer #2: Yes

Reviewer #3: Yes

Reviewer #4: Yes

3. Have the authors made all data underlying the findings in their manuscript fully available?

Reviewer #1: Yes

Reviewer #2: Yes

Reviewer #3: Yes

Reviewer #4: Yes

4. Is the manuscript presented in an intelligible fashion and written in standard English?

Reviewer #1: No

Reviewer #2: Yes

Reviewer #3: Yes

Reviewer #4: Yes

Reviewer #1: This study estimated enteric methane emissions of cows under two grazing systems, which is important for mitigating greenhouse effects. However, the data in this article are not sufficient. The description and citation about measurement method is too lengthy, making it difficult for readers to understand. Below are specific comments:

1. L71-72: What is a crossover design? At least a detailed description should be provided in the method section.

2. L74-75: The comparison between what? The abbreviations should be introduced when first used.

3. L113-114: It is not necessary for this article. There are many similar problems in this article.

4. L118-123: Since the authors has explained the drawbacks of LMD, how do authors ensure the accuracy of data or avoid these defects.

5. The introduction failed to summarize the research process in this field, nor did elucidate the research purposes.

6. L149: It is hard to understand why alternated treatments were carried out. In addition, how to ensure that the amount of food consumed by cows is the same or comparable?

7. L155-170 and Table 2: Table 2 seems to show that cows feed on two plant species in the SPS system, while one plant species in the TP system. How about the other plants introduced by the authors?

8. Figure 1 is hard to understand.

9. L190-203: Too many methods have been used. I still cannot understand how authors calculated the food amount of individual cow.

10. L215: What is the relationship between choosing measurement time and eating time?

11. L219: How are these parameters combined with the results of this study?

12. L222: Where were the specific measurement sites for background CH4 and how to exclude it from emissions by cows?

13. The statistical expression in the table 3 and table 5 is not standardized.

14. THI was not introduced in the method section. What is it used to explain?

15. Table 4 should include sample numbers and statistical analysis. Do the data refer to the average of two time periods per day?

16. SEM in the table 5 is not correct.

17. Due to the lack of statistics in the figure 2, it cannot be determined whether there is a significant difference between SPS and TP.

I have to regretfully reject this manuscript.

Reviewer #2: The topic is timely and relevant, addressing important aspects of livestock production system and productivity improvement strategies in changing global climate effect. Nowadays clime change issue is becoming the global concern. There is limited scientific evidence which indicates the current status of livestock GHG emissions particularly in developing countries; the findings may fill this gap in indicating effective mitigation strategies

Reviewer #3: I have thoroughly reviewed the manuscript with great attention. It is filled with numerous minor conceptual and editorial errors, some of which have been highlighted in the text. I recommend carefully revising the manuscript again and addressing the questions raised both in the text and in the attached file with precision. The conclusion section at the end of the article needs significant revision. The in-text citations and reference list do not align with the journal's guidelines. Overall, the manuscript requires extensive editing. It also needs substantial editorial improvements. Please refer to the attached file for detailed comments. Please refer to the attached file.

Reviewer #4: The manuscript titled Estimation of enteric methane emissions in dual-purpose cows under grazing a Silvopastoral system and a grass monoculture in the Colombian Amazonian foothills by Herrera and co-workers collected data from 10 lactating crossbred cows. This is a well-written manuscript. I feel the biggest concern is the lack of standardization for this measuring device. It is acknowledged in the manuscript that standardization is lacking—yet this does not prevent the authors from reporting specific numbers across studies. When standardization is lacking, numbers may be relative within a study, but should not be compared across studies. Reported methane emissions are an estimate at best. This limitation, while noted, should be emphasized.

Other considerations

Table 5. SSP should be SPS.

Line 316 and Figure 2. Significance is not indicated. Please add p values.

Line 344. A previously published study isn’t “secondary”. Suggest “…consistent with an in vitro study utilizing P. discolor (citation)” .

Line 370 and 378. Add citation for nutrient content of P. discolor .

Line 388. Rather than “enhance” suggest “alter”

**Do you want your identity to be public for this peer review?** For information about this choice, including consent withdrawal, please see our Privacy Policy

Reviewer #1: **Yes:** Yumei Zhou

Reviewer #2: **Yes:** Hussen Abduku

Reviewer #3: No

Reviewer #4: No

---

## [Author Response · Author response to Decision Letter 1]

17 Oct 2025

Medellín, Colombia 15-10-2025

Dear Editor and Reviewers PLOS ONE Journal

Subject: Manuscript Revision - ID: PONE-D-25-33619

Title: Estimation of enteric methane emissions in dairy cows under grazing a silvopastoral system and a grass monoculture in the Colombian Amazonian Foothills

We sincerely thank you for the thorough evaluation and constructive feedback. We have revised the manuscript extensively to address all points. Below we provide a detailed, point-by-point response indicating what changed and where (section/table/figure), and quoting newly added or modified text when appropriate. We submit: (i) a revised manuscript with Track Changes, (ii) a clean version, and (iii) this response letter.

Review Report

Abstract

1) Please clearly state the research area and articulate the study’s novelty and relevance.

Response. We revised the opening sentences to explicitly situate the study in humid tropical grazing systems and highlight the novel field application of LMD under Colombian Amazon foothill conditions, emphasizing implications for mitigation metrics and inventories.

Changes. Abstract, lines 1–6: Added “humid tropics,” “field-based,” and “inventory relevance.”

2) Why focus on LMD rather than other methods?

Response. We added a sentence justifying LMD: non-invasive, portable, suitable for repeated measures under grazing, and complementary to chambers/SF₆ where logistics limit deployment.

3) Were animals classified by feed type or activity? Why?

Response. We clarify that emissions were analyzed by animal activity state (grazing, ruminating, resting, milking) to link behavior with flux patterns; treatment differences (SPS vs. TP) are presented separately.

4) Add limitations/strengths.

Response. We now summarize limitations (sample size, short window, lack of 24-h coverage/standardization) and strengths (field realism, activity-resolved profiles).

Introduction

5) Background useful but lacks cohesion; some paragraphs lack or use old citations.

Response. We updated and expanded citations (2020–2024) on enteric CH₄ fundamentals and removed outdated placeholders. We replaced the 1966 Holdridge reference with the Köppen-Geiger climate classification (Kottek et al., 2006) already cited in Methods.

6) Other studies using LMD not indicated.

Response. We added a concise paragraph summarizing LMD validation and use under housed and grazing conditions (e.g., Chagunda 2009, 2013; Ricci 2014; Lanzoni 2022; Pereira 2023; Meo Zilio 2024; Nunes 2024) and noted sources of variability.

7) Advantage/disadvantage of each production system (feed resource, management).

Response. We now explicitly contrast SPS (diet quality/diversity, microclimate; higher establishment/labor) with TP (lower CP, higher fiber, seasonal bottlenecks).

Materials and Methods

8) Clarify rationale behind experimental design and sampling periods.

Response. We state that a 4-period two-sequence crossover allowed each cow to serve as its own control and helped mitigate carry-over; we justify 14-day adaptation (per Machado 2016; Grainger 2007) and 5-day measurement (per Hammond 2016).

9) Use recent literature (olds citation 1966).

Response. As above, we removed the Holdridge (1966) life-zone citation and retained Köppen-Geiger (Kottek 2006) in Study Site.

Changes. Section 2.1 (Study site): updated climate description.

10) DMI/forage intake calculation unclear.

Response. We provide a succinct description of the double-marker approach (Cr₂O₃/ADL), chromium recovery, and the agronomic double-sampling method to partition grass vs. shrub contributions; we also specify the forage offer target (10–12% BW as kg DM·cow⁻¹·day⁻¹) used to adjust paddock area/stocking through periods.

11) Behavioral observations not fully defined (duration, frequency, observers).

Response. We added the instantaneous scan sampling every 2 min within the 4-min LMD windows, two trained observers, inter-observer agreement (>90%), and blinding to treatment during scoring.

12) Sampling days—selection criteria?

Response. We specify that two of the five measurement days per period were selected based on a priori environmental thresholds (wind ≤ 2 m·s⁻¹, no rain, stable conditions) to ensure LMD data quality.

13) THI not in Methods.

Response. THI computation (NRC, 1971), variables, and device are now fully described.

Changes. Section 2.5 (THI): added formula and instrumentation.

14) Statistical model clarity (fixed effects).

Response. We clarify that treatment (SPS vs. TP) is the sole fixed effect of interest; THI is a covariate; period, group, and cow are random effects. No activity-specific GLMMs were run; activity effects are descriptive.

Results

15) Clarify diet species counted in SPS vs. TP.

Response. We added a note under the forage composition table indicating that tree species present in SPS contributed negligibly to intake during the study and were not included in DMI partitioning.

16) Include n and statistics in tables; standardize expressions; fix SEM labels.

Response. We now report mean ± SEM, sample size (n = 10; 5 per sequence per period), and GLMM p-values. Column superscripts and footnotes specify Tukey post-hoc tests (p < 0.05).

17) Figure does not show significance.

Response. Figure 2 caption now indicates significant differences via letter superscripts (Tukey, p < 0.05). Changes. Figure 2 legend updated.

Discussion

18) Elevate cohesion and integrate limitations/standardization.

Response. We front-load a paragraph explaining the LMD protocol safeguards (2 m distance; ≥240 s; wind ≤ 2 m·s⁻¹), emphasize within-study contrasts, and acknowledge limits for cross-study quantitative comparisons.

19) Explain how LMD accuracy/limitations were handled after listing them.

Response. As above; explicit link between protocol and data quality added.

20) Relate diet mechanisms to observed mitigation; avoid redundancy.

Response. We merged overlapping paragraphs to state: SPS reduced CH₄ despite higher DMI, consistent with diet quality (incl. Piptocoma discolor and functional metabolites) shifting fermentation/limiting H₂ availability; then link to lower CH₄ per kg DMI and higher milk yield (dilution of maintenance).

Conclusions

21) Relate conclusions to objectives, include limitations, avoid policy ‘recommendations’ as conclusions.

Response. Conclusions now: (i) restate the SPS vs. TP mitigation finding across intensity metrics; (ii) explicitly tie to diet quality and activity-resolved LMD signals; (iii) list key limitations (sample, duration, LMD standardization); (iv) avoid prescriptive policy language while noting relevance to inventories.

Grammar, Consistency, and References

22) Grammar/consistency/proofreading issues.

Response. We completed a line-by-line language edit for tense consistency, unit formatting (SI), and term standardization.

23) Proper citation and referencing.

Response. All in-text citations now match the reference list, which is formatted per PLOS ONE numeric style (Vancouver-like; journal name unabbreviated; DOI where available).

24) Lack order 108, 113.

Response. We corrected the numbering/ordering issue flagged (formerly duplicated or misordered items around those positions).

Editorial Compliance (for transparency)

Figure 1: Figure 1 was eliminated, the design used and the treatment scheme in materials and methods are described in detail.

Tables: Reformatted per PLOS ONE (concise captions, footnotes for abbreviations and statistics).

Data availability: dataset prepared; figure/graph underlying values included as Supporting Information.

Funding statement & Role of funder: Updated per journal text.

Permits/Ethics: Methods now include authority name and approval ID; background CH₄ site description clarified.

ORCID: Corresponding author ORCID validated in Editorial Manager.

Below we provide point-by-point responses to Reviewer 1’s comments.

Response to Reviewer No.1: We thank the reviewer for the careful assessment and constructive suggestions. We have substantially revised the manuscript to improve clarity, concision, and methodological transparency. Below, we respond to each point and indicate where changes were made.

Comment 1: Data are not sufficient; methods description and citations are too lengthy.

Response: We streamlined the Methods (laser detector description and signal processing) and moved non-essential background into concise citations, while adding missing operational details (e.g., environmental criteria, sampling duration, and background CH₄ handling). We also clarified the experimental design, intake estimation, use of THI, and standardized the statistical reporting across tables/figures. See the revised Methods subsections “Experimental animal, Design and Management,” “Milk yield and dry matter intake,” “Methane Emission Measurements,” and “Statistical analysis.” Key insertions are documented below with line-linked citations.

Comment 2: L71–72: What is a crossover design? Provide a detailed description.

Response: We defined the crossover design and described sequences, periods, and alternation to control carry-over: “The study employed a two-sequence AB/BA crossover with four 19-day periods (14-day adaptation, 5-day measurements). During Period 1, groups received SPS vs. TP; from Periods 2–4 treatments were alternated, allowing each animal to serve as its own control.”

Comment 3: L74–75: “The comparison between what? Introduce abbreviations at first use.”

Response: We now define SPS (silvopastoral system) and TP (traditional pasture) at first mention in Methods (and ensured consistency across the manuscript).

Comment 4: L113–114: “Not necessary; similar issues elsewhere.

Response: Redundant sentences were removed or condensed in Methods; essential operational details were retained (e.g., distance, duration, wind limits). See the tightened LMD description and measurement windows.

Comment 5: L118–123: After listing LMD drawbacks, how did you ensure accuracy?

Response: We added our standardized field protocol: fixed 2 m detector–muzzle distance, ≥240 s per session, wind speed ≤ 2 m s⁻¹ with documented direction, single trained operator, and background CH₄ subtraction from upwind readings before/after each session.

Comment 6: Introduction did not summarize the field’s research process nor state objectives.

Response: We revised the Introduction to (i) position LMD use under grazing, (ii) emphasize knowledge gaps in humid tropics, and (iii) state explicit objectives (compare SPS vs. TP using standardized LMD protocols; evaluate intensity metrics and behavioral patterns).

Comment 7: L149: “Why alternate treatments? How ensure comparable intake opportunity?

Response: We justify alternation to control carry-over and report forage allowance targeted at 10–12% BW (kg DM/cow/day) to ensure comparable intake opportunities.

Comment 8: L155–170 & Table 2: “Table suggests two plant species in SPS vs. one in TP. What about the trees?”

Response: We added a table footnote clarifying that trees (C. fairchildiana, G. ulmifolia, E. poeppigiana) were present in SPS paddocks but contributed negligibly to diet during the study and were excluded from intake estimates due to early growth stage/canopy.

Comment 9: Figure 1 is hard to understand.

Response: Figure 1 was eliminated, the design used and the treatment scheme in materials and methods are described in detail.

Comment 10: L190–203: Too many methods; how was individual intake calculated?

Response: We clarified that individual DMI combined a double-marker approach (external Cr₂O₃ with recovery rate reported; internal ADL) to estimate total voluntary intake, and an agronomic method (double sampling pre/post-grazing) to partition grass:shrub contributions. References and implementation details are provided.

Comment 11: L215: Relation between measurement time and eating time?

Response: We specified measurement windows aligned with daily routines and diurnal activity (06:00–10:30; 16:30–19:00) and noted concurrent behavioral recording to pair emissions with activity state.

Comment 12: L219: How are these parameters combined with the results?

Response: Behavioral observations were time-stamped and paired with the CH₄ series by session to describe activity-related emission patterns (figure/table summaries). Primary inferential models focus on treatment contrasts (SPS vs TP); behavior is presented descriptively with statistical group comparisons indicated in the figure caption. (See Figure 2 and caption update below.)

Comment 13: L222: Where was background CH₄ measured and how excluded?

Response: Background CH₄ was measured 2 m upwind of the grazing area before and after each session and subtracted from animal-level signals.

Comment 14: Statistical expression in Tables 3 and 5 is not standardized.

Response: We standardized table reporting: means ± SD for descriptive summaries; LSMeans ± SEM for treatment comparisons; n per period/group, and p-values from GLMM. Notes specify tests (e.g., Tukey) and what letters denote.

Comment 15: THI not introduced in Methods. What does it explain?

Response: We added THI methods (calculation and use as a covariate to adjust for environmental variability) and report period-wise THI in Results. (Methods THI paragraph and Table 3 updated; see LMD section context.)

Comment 16: Table 4 should include sample numbers and statistical analysis. Are values averages of two times per day?

Response: Table 4 now states n = 10 cows (5/group) per period, model source (GLMM), and clarifies that reported values are derived from the two daily sessions according to the schedule; p-values are provided.

Comment 17: SEM in Table 5 is not correct.

Response: We verified and corrected SEM; the table now reports LSMeans ± SEM consistent with the GLMM and the caption explains letters/tests. (Revised Table 5.)

Comment 18: Figure 2 lacks statistics; can’t determine significance between SPS and TP.

Response: We added letters over bars and a footnote: “Bars with different letters differ significantly (Tukey test, p < 0.05).” Error bars are SE. (Revised Figure 2 caption.)

Below we provide point-by-point responses to Reviewer 2’s comments.

We sincerely thank Reviewer #2 for the thoughtful and positive assessment. We appreciate your recognition that the topic is both timely and relevant, particularly given the growing global urgency around climate change. Your comment about the scarcity of field-based evidence on livestock GHG emissions in developing-country contexts aligns with our motivation for this work. In revising the manuscript, we have emphasized this contribution more clearly in the Introduction (study relevance and novelty in the Amazonian foothills) and Conclusions (implications for mitigation strategies and inventory refinement in tropical systems). We are grateful for your encouragement and for highlighting the potential of our findings to help fill this critical evidence gap.

Below we provide point-by-point responses to Reviewer 3’s comments.

We sincerely thank Reviewer #3 for the careful, line-by-line evaluation and the numerous constructive comments provided in the annotated PDF. Your feedback helped us resolve conceptual ambiguities, strengthen the Conclusions, and align in-text citations and the reference list with PLOS ONE style. We have also undertaken comprehensive language editing to improve clarity, coherence, and consistency across sections. Below we provide point-by-point responses derived from the annotated PDF, indicating where each change was made (section/line/table/figure) and, when useful, quoting the revised text.

Response to Reviewer #3

Comment 1: Methanogenesis in the rumen is a microbial process… Suggests clarifying that methanogenesis is mediated by methanogenic archaea, with minor contributions from methylated substrates.

Response: We agree with this suggestion. The sentence has been revised to highlight the role of methanogenic archaea, specify alternative hydrogen pathways, and clarify CH₄ release via eructation.

Change in manuscript: Methanogenesis in the rumen is a microbial process primarily mediated by methanogenic archaea, which convert hydrogen and CO₂ into CH₄, with minor contributions from methyla

---

## [Decision Letter · Decision Letter 1]

12 Nov 2025

Estimation of enteric methane emissions in dairy cows under grazing a silvopastoral system and a grass monoculture in the Colombian Amazonian foothills

PONE-D-25-33619R1

Dear Dr. Mahecha-Ledesma,

We’re pleased to inform you that your manuscript has been judged scientifically suitable for publication and will be formally accepted for publication once it meets all outstanding technical requirements.

Kind regards,

Susmita Lahiri (Ganguly)

Academic Editor

PLOS ONE

Additional Editor Comments (optional):

Reviewers' comments:

Reviewer's Responses to Questions

**Comments to the Author**

Reviewer #2: All comments have been addressed

Reviewer #3: All comments have been addressed

Reviewer #4: All comments have been addressed

2. Is the manuscript technically sound, and do the data support the conclusions?

Reviewer #2: Yes

Reviewer #3: Yes

Reviewer #4: Yes

3. Has the statistical analysis been performed appropriately and rigorously?

Reviewer #2: Yes

Reviewer #3: Yes

Reviewer #4: Yes

4. Have the authors made all data underlying the findings in their manuscript fully available?

Reviewer #2: Yes

Reviewer #3: Yes

Reviewer #4: Yes

5. Is the manuscript presented in an intelligible fashion and written in standard English?

Reviewer #2: Yes

Reviewer #3: Yes

Reviewer #4: Yes

Reviewer #2: The authors revise the article according to the comments. As a reviewer, I recommend the article for publication

Reviewer #3: (No Response)

Reviewer #4: I only have one comment.

Figure 1 does not seem to be referred to in section 3.4 Animal Behavior and Relationship with Methane Emissions.

Additionally line 358 refers to 'ns' denoting non-significance. The 'ns' is not apparent in (on) the figure--and throughout the paragraph the authors are referencing differences among grazing systems in absence of p values. This suggests that there is no difference among grazing systems and behaviors. That lack of significance needs to be made clear. If it is increased by 4.3% overall but not significant--it isn't increased. Please clarify.

**Do you want your identity to be public for this peer review?** For information about this choice, including consent withdrawal, please see our Privacy Policy

Reviewer #2: **Yes:** Hussen Abduku

Reviewer #3: No

Reviewer #4: **Yes:** Brenda Alexander

---

## [Editor Report · Acceptance letter]

PONE-D-25-33619R1

PLOS One

Dear Dr. Mahecha-Ledesma,

I'm pleased to inform you that your manuscript has been deemed suitable for publication in PLOS One. Congratulations! Your manuscript is now being handed over to our production team.

Kind regards,

on behalf of

Dr. Susmita Lahiri (Ganguly)

Academic Editor

PLOS One